# Unusual topological polar texture in moiré ferroelectrics

Yuhao Li [1,2], Yuanhao Wei[3], Ruiping Guo [4,5], Yifei Wang [4], Hanhao Zhang[1], Takashi Taniguchi [6], Kenji Watanabe [7], Yan Shi [3], Yi Shi [1] ✉, Chong Wang [4] ✉ & Zaiyao Fei [1,2] ✉

Topological polar textures in ferroelectrics have attracted significant interest for their potential applications in energy-efficient and high-density data storage and processing. Among these, polar merons and antimerons are predicted in strained and twisted bilayers of inversion symmetry broken systems. However, experimental observation of these polar textures within twisted two-dimensional van der Waals materials remains elusive. Here, we utilize vector piezoresponse force microscopy to reconstruct the polarization fields in R-type marginally twisted hexagonal boron nitride. We observe alternating out-of-plane polarizations at domain regions and in-plane vortex-like polarization patterns along domain walls, indicative of a network of polar merons and antimerons. Notably, the out-of-plane polarization exhibits three polarity reversals across a domain wall. Similar polar textures are identified in marginally twisted WSe$_2$ and MoSe$_2$ homobilayers. Our theoretical simulations attribute these unusual polarization reversals near the domain walls to the competition between moiré ferroelectricity and piezoelectricity. These results provide the experimental evidence of complex polar textures in moiré ferroelectrics, which may offer additional insights into the electronic band topology in twisted transition metal dichalcogenides.

The intricate interplay of various degrees of freedom, such as lattice, charge, spin and orbit in condensed matter can give rise to diverse topological structures in real space[1–3]. For instance, the Dzyaloshinsky-Moriya spin-orbit interaction in magnetic thin films can foster topological structures like magnetic skyrmions[4,5] (winding number $N = \pm 1$) and merons[6] ($N = \pm 1/2$). A meron can be viewed as half a skyrmion with spins in the core area pointing out-of-plane (OOP), gradually evolving into in-plane (IP) vortex-like pattern at the boundary. The electric counterpart, topological polar textures, have

also been demonstrated but only in a few ferroelectric systems, such as oxide superlattices[2,7–11].

Recently, interfacial ferroelectricity has been discovered in non-centrosymmetric two-dimensional (2D) van der Waals materials through specific stacking arrangements or interlayer twisting[12–17]. This new ferroelectric system provides unique opportunities for exploring and manipulating exotic electronic states[18–26] as well as topological polar textures. In particular, for R-type twisted hexagonal boron nitride (hBN) and transition metal dichalcogenide (TMDC)

[1]National Laboratory of Solid-State Microstructures, School of Electronic Science and Engineering and Collaborative Innovation Center of Advanced Microstructures, Nanjing University, Nanjing, Jiangsu, China. [2]National Key Laboratory of Spintronics, Nanjing University, Suzhou, Jiangsu, China. [3]State Key Laboratory of Mechanics and Control for Aerospace Structures, Nanjing University of Aeronautics and Astronautics, Nanjing, Jiangsu, China. [4]State Key Laboratory of Low Dimensional Quantum Physics and Department of Physics, Tsinghua University, Beijing, China. [5]Institute for Advanced Study, Tsinghua University, Beijing, China. [6]Research Center for Materials Nanoarchitectonics, National Institute for Materials Science, Tsukuba, Japan. [7]Research Center for Electronic and Optical Materials, National Institute for Materials Science, Tsukuba, Japan. ✉e-mail: yshi@nju.edu.cn; chongwang@mail.tsinghua.edu.cn; zyfei@nju.edu.cn

homobilayers, alternating OOP ferroelectric polarization emerges due to inversion symmetry breaking, a phenomenon known as moiré ferroelectricity[27]. On the other hand, since the individual layer lacks inversion symmetry, the piezoelectric effect may also contribute to electric polarization in moiré ferroelectrics, especially near the domain walls (DWs) where strain is confined to. In experiment, OOP and IP polarizations have been observed in moiré ferroelectrics within the domains and near the DWs[28,29], respectively. However, the origin of the electric polarization and its evolution in these systems remain poorly understood.

In this study, we methodically investigate the OOP and IP electromechanical responses of marginally twisted double trilayer hBN, bilayer $WSe_2$ and bilayer $MoSe_2$ by vector piezoresponse force microscopy (PFM). Within the moiré domains, only OOP polarizations are detected, exhibiting opposite polarities in adjacent domains, consistent with moiré ferroelectricity. Near the DWs, both OOP and IP polarizations are observed. The IP components are found to align along the DWs through angular-dependent lateral PFM (LPFM) measurement, forming vortex-like structures. Of particular interest is the distribution of OOP components near the DWs. By suppressing the cantilever buckling effect induced by IP polarization during vertical PFM (VPFM) measurement, we obtain the intrinsic OOP polarization field near the DWs. The OOP polarization in twisted hBN reverses three times across a DW, instead of a monotonical transition. The unusual polar textures are also observed in twisted $WSe_2$ and $MoSe_2$ homobilayers, revealing a competitive interplay between ferroelectricity and piezoelectricity in moiré ferroelectrics.

## Results

We first look at twisted hBN samples with a tiny twist angle of below 0.05°, which are comprised of exposed twisted hBN sitting atop thin graphite substrates (see "Methods" and Figs. S1, S2 for fabrication

details). Figure 1a shows the atomic arrangements for AB, BA and saddle-point (SP) stackings of a R-type twisted bilayer hBN moiré superlattices (Fig. 1b). The yellow and purple arrows indicate the orientations of electric dipoles at the interface of AB and BA stackings, respectively. Due to an in-plane two-fold rotational symmetry, the SP stacking has vanishing OOP electric dipoles.

Figure 1c shows the Kelvin probe force microscopy (KPFM) image of a twisted double trilayer hBN sample (thBN1), which gives negative and positive surface potential for the commensurate AB and BA stacking domains, respectively. While KPFM effectively images the surface potential, it primarily reflects the OOP polarization and cannot measure the IP polarization near the DWs. Furthermore, the DWs separating the AB and BA stacking domains are on the order of 10 nm wide due to atomic reconstruction, which exceeds the resolution limit of KPFM. To reconstruct the full polarization field with higher resolution, we switch to vector PFM for the rest of this study (see Fig. S3 for experimental comparison of the distinct spatial resolutions achieved by in-situ KPFM and PFM measurements on thBN2). During the measurements, the cantilevers are always aligned along the horizonal axis, unless otherwise specified. Typically, the VPFM signal (vertical deformation) is associated with the OOP component of the electric polarization, whereas of the LPFM signal (lateral deformation) is associated with the IP component.

Figure 1d, e show the decoupled VPFM phase and amplitude images of the same sample (see Fig. S4a, b and Supplementary Note 1 for the raw images and discussions of the decoupling processes). A 180° phase contrast is observed between the AB and BA stacking domains, while their amplitudes are comparable. LPFM images of the same area show vanishing IP signals within the AB and BA domains (Fig. S5). These suggest that the adjacent stacking domains are oppositely polarized OOP, consistent with the KPFM image in Fig. 1c. Surprisingly, for different moiré DWs, the VPFM signals seem to

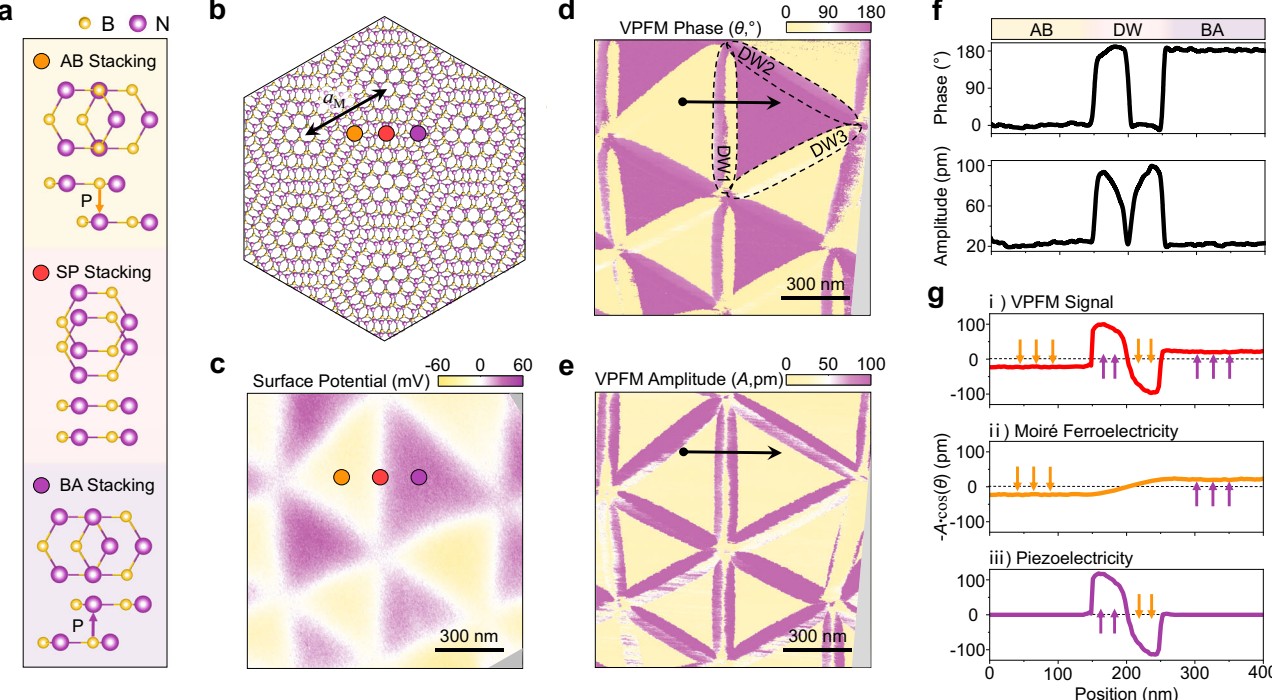

**Fig. 1 | KPFM and VPFM measurements on a marginally twisted hBN sample (thBN1). a** Schematics of AB, BA and SP stacking domains in a twisted hBN moiré superlattices. **b** Schematic of a twisted hBN moiré superlattices. The relation between twist angles ($\theta_T$) and measured moiré periodicity ($a_M$) of twisted hBN moiré superlattice is given by $a_M = a/(2\sin(\theta_T/2))$, where $a$ is the lattice constant of hBN. **c**–**e** Surface potential, VPFM phase ($\theta$) and VPFM amplitude ($A$) images of a

twisted double trilayer hBN with twist angle from 0.02° to 0.027°. **f** Line cut of VPFM phase and amplitude along the black arrowed line in (**d**, **e**). **g** Line cuts of (i) measured in-phase VPFM signal ($-A\cdot\cos(\theta)$), (ii) in-phase VPFM signal due to moiré ferroelectricity and (iii) in-phase VPFM signal due to piezoelectricity along the black arrowed lines in (**d**, **e**).

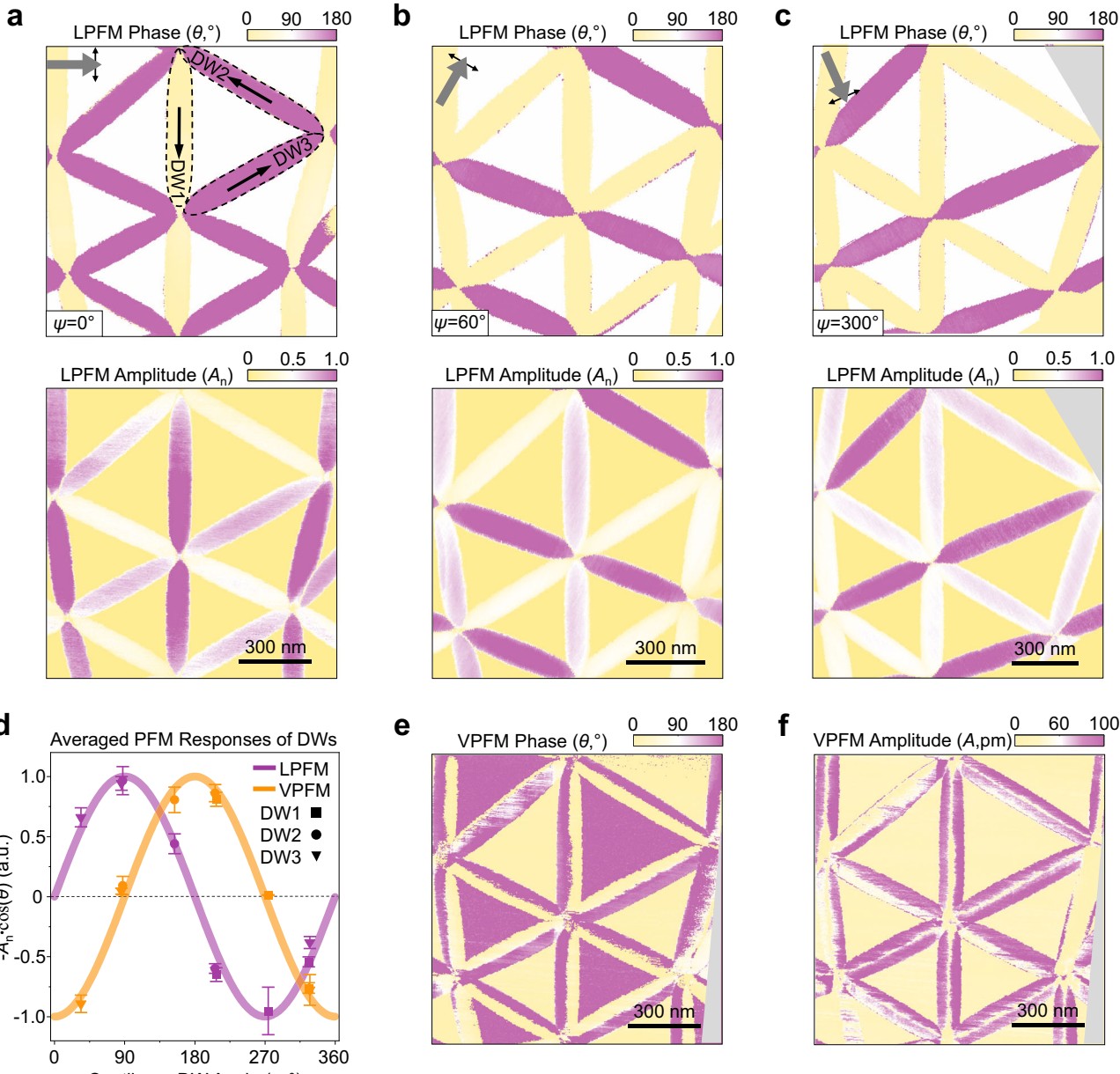

**Fig. 2 | Angle-dependent LPFM measurements and buckling effect-free VPFM images. a–c** LPFM phase and normalized LPFM amplitude images obtained while rotating the sample. The cantilever-sample angle ($\psi$) are 0°, 60° and 300°, respectively. **d** The normalized in-phase LPFM and VPFM signal ($-A_n \cdot \cos(\theta)$) as a function of the cantilever-DW angle ($\varphi$). The VPFM signals are shifted by 90° with respect to those of LPFM for all DWs. **e, f** Buckling effect-free VPFM phase and amplitude images of Fig. 1d, e.

depend on the relative orientation between the cantilever and the DW, which is also illustrated in Fig. 2d and Supplementary Note 2. This contradicts the expected three-fold rotational symmetry for the OOP polarization. Moreover, across DWs that are nearly perpendicular to the cantilever, the in-phase VPFM signals, $-A \cdot \cos(\theta)$ (the minus sign aligns the signal with the polarization direction), exhibit extra polarity reversals, as exampled in the line cut of DW1 (Fig. 1f).

To understand the DW-specific VPFM signals, we perform a series of vector PFM measurements while rotating the sample. The decoupled LPFM phase and amplitude images are shown in Fig. 2a–c, where $\psi$ measures the rotation of the sample. For clearer visualization, we rotate the cantilever, as indicated by the arrows on the top left insets representing the cantilever's relative positions. The normalized in-phase LPFM signal, $-A_n \cdot \cos(\theta)$, as a function of the cantilever-DW angle

($\varphi$) is plotted in Fig. 2d (see Fig. S6 for the normalization process), which can be fitted to a sine curve (purple line). Since the LPFM detects the component of the IP polarization perpendicular to the cantilever[30], the sine fit implies that the IP polarization near the DWs are along the DWs, thus forming clockwise and anti-clockwise vortex-like patterns (see the black arrows in the top panel of Fig. 2a and the color map of Fig. S7 for the reconstructed full IP polarization field), similar to the results obtained in graphene and TMDC moiré patterns[31–35]. On the other hand, the normalized in-phase VPFM signal after averaging on each DW can be fitted into a cosine curve (orange line, the angle-dependent VPFM images can be found in Fig. S8). The observed 90° shift suggests that the averaged VPFM signals can be associate with a component of the IP polarization that lies along the cantilever direction, which is known as IP polarization induced cantilever buckling effect[36,37]. Additional discussion of the buckling effect can be found in

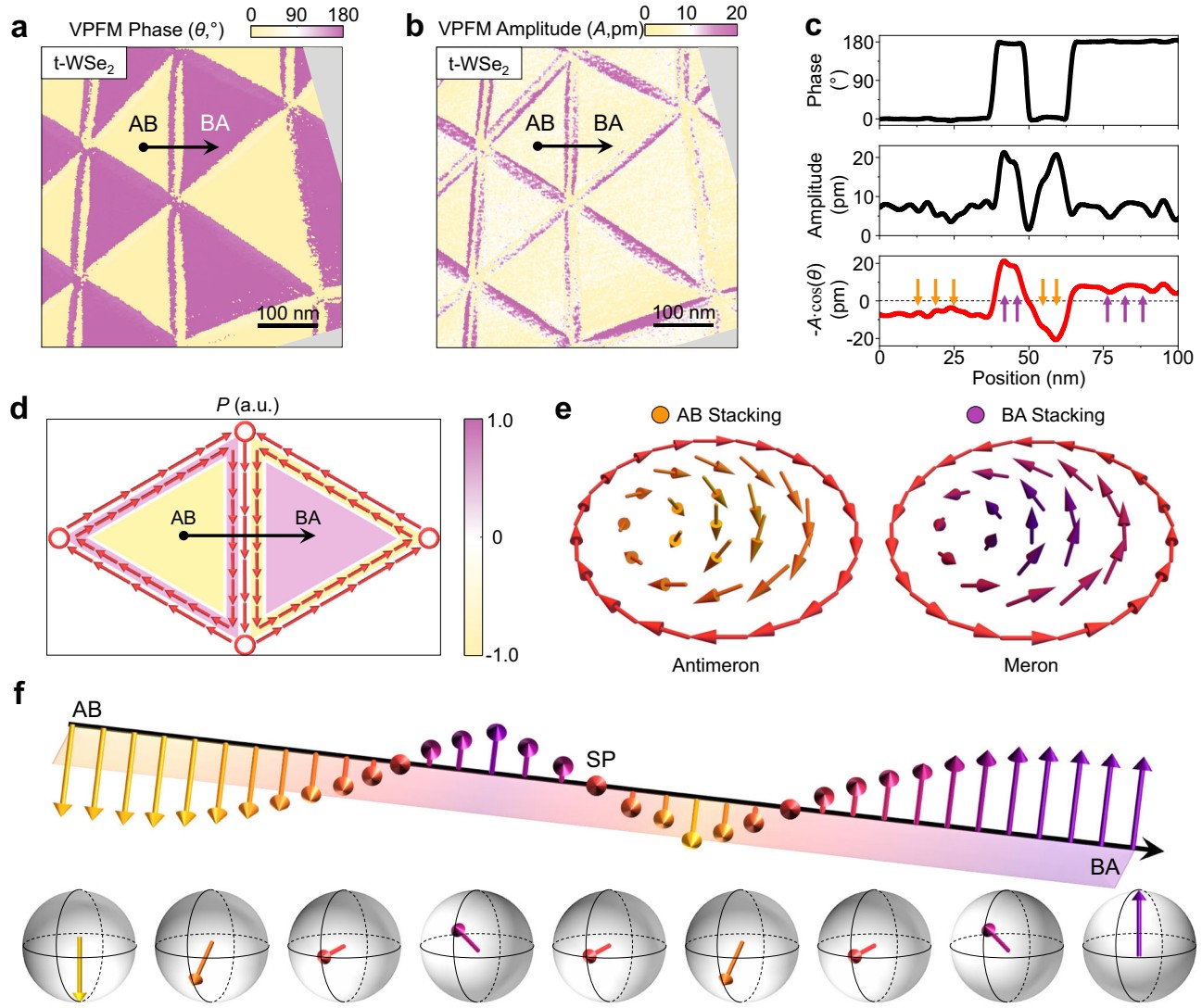

**Fig. 3 | VPFM measurement of a twisted bilayer WSe₂ sample and evolution of the electric polarization.** VPFM phase (**a**) and amplitude (**b**) images of a twisted bilayer WSe₂ sample with twist angle from 0.049° to 0.064°. **c** VPFM signals along the black arrowed lines in (**a**, **b**). **d** Schematic of the polar texture of the twisted hBN in Figs. 1, 2. **e** Polar meron and antimeron structures. **f** Evolution of the electric polarization along the black arrowed line in (**d**) and the relative orientations on a ferroelectric Bloch sphere.

Fig. S9 and Supplementary Note 2. In Figs. S10–S12, we provide angle-dependent vector PFM measurements on another twisted hBN sample (thBN3), which reveals similar results.

In contrast to existing approaches for suppressing cantilever buckling effects[36,37], the alignment of IP polarization along DW directions in our system enables straightforward suppression by orienting the cantilever perpendicular to the DWs, as for the highlighted DW (DW1) in Fig. 1d, e. VPFM images of the same sample while DW2 and DW3 are aligned perpendicular to the cantilever are provided in Fig. S8. In this case, the in-phase VPFM signal in Fig. 1g(i) is assigned to the OOP polarization. Three polarity reversals are observed across the DW for the OOP polarization, marking the key finding of this work. At the center of the DW (SP stacking), the OOP polarization vanishes, consistent with the symmetry analysis above. We also plot the moiré ferroelectric contribution to the VPFM signal in Fig. 1g(ii)[38,39]. The remaining VPFM signal (Fig. 1g(iii)) is attributed to piezoelectric effect[40], as corroborated by the DFT calculations presented later. We note the piezoelectric contribution to the polarization has the opposite sign to that of the moiré ferroelectricity, indicating a competitive interplay between the two within the DWs. We also realize the relative amplitude of the buckling effect varies with several factors, such as the

position of the laser spot (see Supplementary Note 2 and Fig. S13 (thBN4) for more information).

Alternatively, the intrinsic OOP polarization can be extrapolated from symmetry analysis in this particular system. As AB and BA stackings are related by an IP two-fold rotational operation, the OOP polarization is antisymmetric with respect to the SP stacking, while the IP polarization is symmetric. Figure 2e, f show the antisymmetrized VPFM images, where three polarity reversals are observed across nearly all DWs, restoring the three-fold rotational symmetry.

We also perform vector PFM measurements on R-type marginally twisted WSe₂ and MoSe₂ homobilayers. Similar to twisted hBN, the commensurate AB and BA stacking domains are also polarized in the OOP directions, whereas the SP stacking domains are nonpolar. Figure 3a, b show VPFM images of a twisted bilayer WSe₂ sample (twist angle -0.06°). Clearly, the AB and BA stacking domains also exhibit the expected 180° phase contrast, with VPFM amplitudes of ~7 pm, nearly one third of that observed in twisted hBN. Near the DWs, the features are less pronounced but still distinguishable due to the reduced piezoelectric effect. Three polarity reversals are observed in the VPFM signal across vertical DWs (Fig. 3c), qualitatively consistent with that in twisted hBN. The IP polarizations are also found to align along the DWs

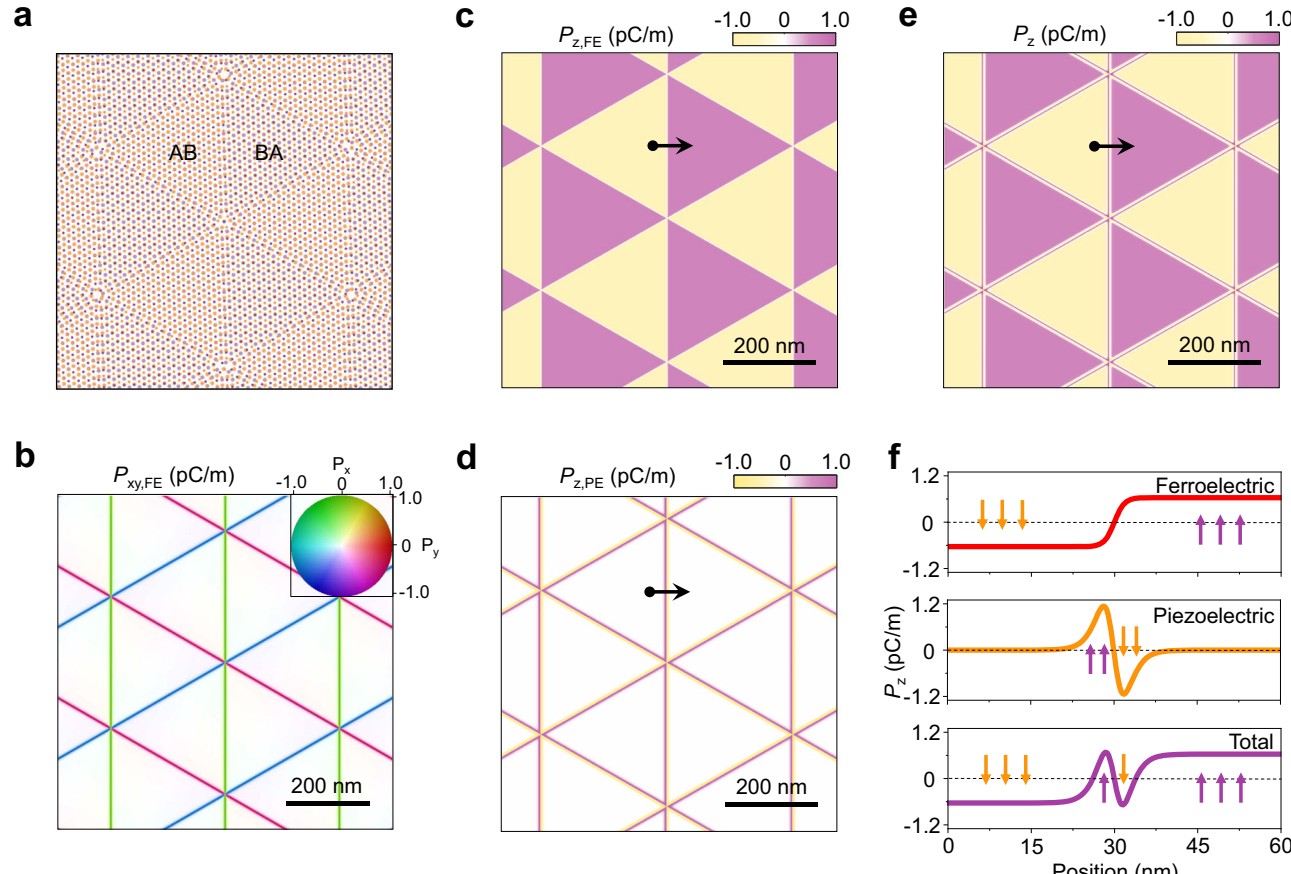

**Fig. 4 | Competition between moiré ferroelectricity and piezoelectricity in a 0.06° twisted bilayer WSe$_2$ moiré superlattice. a** Atomic structure of a relaxed twisted bilayer WSe$_2$ moiré superlattice. **b** Calculated IP polarization map of twisted bilayer WSe$_2$ moiré superlattices. **c**–**e** Calculated OOP ferroelectric polarization, OOP piezoelectric polarization and total OOP polarization maps of twisted bilayer WSe$_2$ moiré superlattices. **f** Line cuts of the OOP polarizations in (**c**–**e**).

as in twisted hBN. PFM results of a 0.09° twisted MoSe$_2$ sample are provided in Fig. S14.

With collected information of both IP and OOP polarizations, we sketch the polar texture of twisted hBN or TMDC homobilayer in Fig. 3d. The yellow and purple shaded triangles denote the commensurate AB and BA stacking domains, where the polarization is purely OOP. In the DW regions, IP polarization emerges, as indicated by the red arrows. Across the DW (denoted by the black arrowed line), the OOP polarization first reverses, reaching a peak value before decreasing in amplitude and vanishing at the DW center. The OOP polarization profile on the other side of the DW is an antisymmetric copy. The polar texture across the DW can also be visualized on a ferroelectric Bloch sphere, as shown in Fig. 3f. For AB stacking, the polarization vector points to the south pole. Across the DW, it rotates to equator first and tilts toward the north pole during OOP polarization reversal. It then returns to the equator at the DW center, where the OOP polarization vanishes.

To illustrate the competition between moiré ferroelectricity and piezoelectricity in twisted hBN or TMDCs, we conducted relaxation simulations by elastic-theory-based continuum model for twisted bilayer WSe$_2$ at a twist angle of 0.06° (See "Methods" for details). The relaxed structure, IP and OOP polarization maps are shown in Fig. 4. After relaxation, large domains with uniform AB and BA stackings emerge, as schematically depicted in Fig. 4a. Moiré ferroelectricity generates both IP polarization along the DWs and OOP polarization within the AB and BA domains (Fig. 4b, c). Within the DW regions, piezoelectricity induces significant OOP polarization (Fig. 4d), opposite in sign to the ferroelectric polarization. Consequently, the

ferroelectric OOP polarization dominates in the domains, while in the DWs, the stronger piezoelectric OOP polarization reverses the sign of the polarization across the DWs (Fig. 4e, f). We also calculate the winding numbers of the unit polarization field for the extended BA and AB domains (including half of the domain walls), which are

$$N = \frac{1}{4\pi} \int \mathbf{P} \left( \partial_x \mathbf{P} \times \partial_y \mathbf{P} \right) ds = \pm 1/2 \qquad (1)$$

This suggests they are polar merons and antimerons, as schematically shown in Fig. 3e.

## Discussion

The unusual polar textures we observed in marginally twisted hBN and TMDC moiré superlattices remain robust across a range of twist angles. Figures S15–S17 present PFM results of two nonuniformly twisted hBN samples (thBN2 and thBN5). While the twist angle varies from 0.014° to 0.375°, the meron-like polarization distribution and multiple OOP polarization reversals persist. These features are reminiscent of the polarization charge distributions in 1.25° twisted bilayer MoTe$_2$ or WSe$_2$ moiré superlattice[40]. Limited by the spatial resolution of PFM (~10 nm) transitions of moiré potential maxima at larger twist angles are unfortunately beyond the reach of this technique. Future studies employing more advanced imaging techniques, such as high-resolution transmission electron microscopy or scanning tunneling microscopy could provide a more comprehensive understanding of

the correlation between real-space polar textures and the band topology in these systems.

Although these universal features appear across different moiré systems, their specific manifestations are influenced by intrinsic material properties. Twisted hBN exhibits stronger interlayer charge transfer and piezoelectric response due to its ionic character, while twisted $WSe_2$ and $MoSe_2$ show more modest charge redistribution but stronger band structure modulation through orbital hybridizations. This difference leads to hBN's more pronounced electrostatic potential variations versus TMDCs' stronger coupling to electronic states.

Beyond twist angle and material-specific properties, the polar textures in moiré superlattices may also be influenced by intrinsic and extrinsic defects. For instance, point defects and grain boundaries can act as pinning sites for electric polarizations[41,42], creating localized dipole fluctuations and altering the switching dynamics. Another example is interlayer dislocations in 2D materials and their heterostructures, often faintly visible in topography, may alter the strain fields and electrostatic landscapes of the moiré pattern[43], thereby modifying the polar textures and electronic properties.

Overall, our study presents the experimental evidence of complex topological polar textures in marginally twisted hBN and TMDC homobilayers, highlighting the unique interplay between moiré ferroelectricity and piezoelectricity. The ability to modulate the polar texture through stacking configurations[44-46], twist angle, or interlayer sliding offers exciting possibilities for engineering topological and electronic states. This opens pathways for integrating moiré ferroelectrics into nanoscale devices, including topological memory elements and reconfigurable logic gates.

## Methods

### Sample fabrication
High quality highly oriented pyrolytic graphite, $WSe_2$ and $MoSe_2$ crystals are sourced from HQ Graphene. Atomically thin flakes were obtained by mechanical exfoliation, with thicknesses initially identified through optical contrasts and subsequently confirmed by AFM. All measured devices (including 5 twisted hBN (thBN1-5), 2 twisted $WSe_2$ and 2 twisted $MoSe_2$ devices) were fabricated using a modified "tear-and-stack" method[47]. Detailed stacking procedures, typical optical images of twisted hBN and TMDCs are presented in Figs. S1, S2. Following stacking, we performed large-scale LPFM scans to identify regions with clean interfaces (from the topography) and desired twist angles. Twist angles ($\theta_T$) were calculated from the measured moiré periodicity ($a_M$) in the LPFM images, following the relation $a_M = a/(2\sin(\theta_T/2))$, where $a$ is the lattice constant of the constituent 2D material. For moiré patterns exhibiting clearly non-equilateral triangular morphologies, we consider the effect of uniaxial heterostrain to determine the twist angle[48]. Selected areas are further refined via AFM tip cleaning to minimize surface contaminants before detailed KPFM and vector PFM studies.

### Atomic force microscopy measurements
Three AFM modes were adopted in this work, i.e., VPFM, LPFM, and KPFM, all performed on an Oxford Instruments Asylum Research MFP-3D Origin AFM at ambient conditions with a relative humidity of ~35%. ASYELEC-01-R2 probes coated with 5-nm Ti and 20-nm Ir were used in all modes with a spring constant of 2.8 N/m. Typical free resonance frequency is ~75 kHz. VPFM and LPFM contact resonance frequency are ~300 kHz and ~780 kHz, respectively. Throughout the PFM measurements, the tip-sample contact force was maintained below 10 nN to minimize mechanical perturbation of the sample. In KPFM measurement, the dual pass amplitude modulation is employed to capture the topography and surface potential difference between tip and sample, respectively. More VPFM and KPFM results of twisted hBN (thBN2) under different relative humidities can be found in Figs. S18, S19. Figure S20 shows similar VPFM results of twisted hBN under different

applied normal forces. To minimize the potential shift, a zero-order flattening process is applied to the raw surface potential data.

### Data processing
The in-phase LPFM and VPFM signals at DWs are collected and Gaussian fittings are performed on them. The mean values and standard deviations of the fittings for LPFM correspond to the projected IP deformations and corresponding error bars, and the VPFM results are the buckling effect of IP deformations. By averaging the VPFM signal for each DW, the contribution of $P_z$ is excluded, leaving only the contribution of IP polarization induced buckling effect.

### Theoretical modeling of local polarization
To simulate the structural relaxation and piezoelectric polarization shown in Fig. 4, we employ an elastic-theory-based method described in ref. 49. This approach decomposes the total energy into elastic and stacking energies, optimizing the continuum atomic displacement field to minimize the total energy. For the elastic energy calculations, we use the bulk and shear modulus of monolayer $WSe_2$ in ref. 50,51. The stacking energy is characterized using the generalized stacking fault energy for parallel bilayer $WSe_2$ from ref. 52. From the relaxation results, we separately compute the piezoelectric polarization and the ferroelectric polarization. The piezoelectric charge density is computed with the piezoelectric coefficient from ref. 53,54. The piezoelectric charge density is assumed to be completely accounted for by the piezoelectric OOP polarization. The ferroelectric polarization pattern in the real space is obtained by mapping the real space to different stacking configurations. By symmetry analysis, up to the first harmonics, the IP and OOP ferroelectric polarization for different stacking configurations have the form

$$P_{xy,FE} = c_{xy} \cdot \begin{pmatrix} 2\sqrt{3}\sin\left(\frac{\sqrt{3}x}{2}\right)\sin\left(\frac{y}{2}\right) \\ 2\cos(y) - 2\cos\left(\frac{\sqrt{3}x}{2}\right)\cos\left(\frac{y}{2}\right) \end{pmatrix}, \qquad (2)$$

$$P_{z,FE} = c_z \cdot \left(2\sin(y) - 4\cos\left(\frac{\sqrt{3}x}{2}\right)\sin\left(\frac{y}{2}\right)\right), \qquad (3)$$

where the lattice vectors are scaled to be $2\pi(\frac{1}{2}, \frac{\sqrt{3}}{2})$ and $2\pi(-\frac{1}{2}, \frac{\sqrt{3}}{2})$. Based on the modern theory of polarization[55], we compute the polarization by density functional theory (DFT) and the coefficients are fitted to be $c_{xy} = 0.265$ pC/m, $c_z = 0.061$ pC/m.

### Density functional theory calculations
DFT calculations were performed using the VASP package[56], employing the projector augmented wave pseudopotential[57,58] and the Perdew-Burke-Ernzerhof (PBE) exchange-correlation functional[59]. The relaxation calculations were performed using a $19 \times 19 \times 1$ Gamma-centered k-point grid, an energy cutoff of 800 eV, the DFT-ulg method[60] for van der Waals correction. The convergence criteria for atomic forces in structural relaxation is $10^{-2}$ eV/Å.

## Data availability
The data that support the findings of this study are available on Figshare at https://figshare.com/s/068b98a1138f6f3fe77b?file=53931797. Additional data are available from the corresponding author (Z.F.) upon reasonable request.

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

## Acknowledgements

This work is supported by the National Key Research and Development Program of China (2021YFA0715600), National Natural Science Foundation of China (12274222 and 12404217), the National Science Foundation of Jiangsu Province (BK20220756) and the Fundamental Research Funds for the Central Universities (2024300420). C.W., R.G. and Y.W. are supported by the startup grant from Tsinghua University. C.W., R.G. and Y.W. acknowledge the discussion with Qiyun Xu and Bozhong Zhang. K.W. and T.T. acknowledge support from the JSPS KAKENHI (21H05233 and 23H02052), the CREST (JPMJCR24A5), JST and World Premier International Research Center Initiative (WPI), MEXT, Japan.

## Author contributions

Z.F. and Yi Shi supervised the project. H.Z., Y.W. and Y.L. fabricated the devices. Y.L.,Y.W. and Yan Shi performed the AFM measurements, R.G., Y.W. and C.W. performed the theoretical modeling. T.T. and K.W. provided the bulk BN crystals. Z.F., Y.L., C.W. and Y.W. analyzed the data. Z.F., Y.L., and C.W. wrote the paper with inputs from all authors.

## Competing interests

The authors declare no competing interests.
