## [Transparent Peer Review file · Nature Communications]

Unusual topological polar texture in moiré ferroelectrics

Corresponding Author: Professor Zaiyao Fei

Version 1:

Reviewer comments:

Reviewer #1

(Remarks to the Author)

This is a beautiful article combining experiment and theory on the polar states of marginally rotated bilayers.

The noteworthy result is the discussion of piezoelectric effects, which are added to the ongoing discussion of topology provided by Bennett and coworkers (reference 36 on the manuscript); besides, while Reference 36 (a computational result) argues for a dipole moment slowly varying across Ab and BA domains, this work demonstrates that the topological polar texture is confined to joining sections of domain walls. It is also worth noting that the authors cross-check their assertions on three different homobilayers.

The work will be of significance to the community working on topological properties of moire materials. Perhaps one omission from the authors is the work of Rebeca Engelke and coworkers (Phys. Rev. B 107, 125413 (2023)), where the topology of the domain walls is firmly established by experiment and theory as well. That team also has a variational model to describe distortions (Journal of Elasticity 154, 443 (2023)). I wonder if the authors can link the shading observed on their figure 2a (second row of results) to the color scheme shown by Engelke and coworkers in the PRB paper mentioned above.

The work does support the conclusions and claims; the authors do a careful job in describing the experimental technique, which integrates out-of-plane and in-plane polar capabilities. No flaws are identified. The battery of results, including a symmetry analysis, do indicate clever ways to separate piezo responses from ferroelectric ones. Perhaps an outstanding question is whether the force microscope tip contributes to the observed piezoelectric strain.

Having said the above, the article can be followed very clearly, and it is my impression that its descriptions are very detailed so as to be reproducible.

Reviewer #2

(Remarks to the Author)

Authors describe an experimental study polar texture in moiré ferroelectrics whereas some simulations are provided with the aim to complement the experimental data. The topic looks properly inserted into the existing literature. The experimental setup seems properly detailed. The results looks properly presented and discussed. The results provide new physical insights into the electronic band topology in twisted transition metal dichalcogenides (TMDs). However, the electronic properties of moiré TMDs are largely studied and many works focus on specific aspects which are suitable for more specialistic journals, as it is the case of the present manuscript. Finally, the lack of quantum mechanical calculations makes the manuscript unsuitable for publication in the Nature Communication journal. In fact, quantum mechanical calculations are necessary to provide a complete insight into the physical phenomena described in the manuscript; in fact, such kind of calculations are commonly employed in scientific articles regarding electronic properties of moiré van der Waals heterostructures.

Therefore, I cannot recommend the publication of the manuscript in the Nature Communication journal. I suggest authors to submit the manuscript to a more specialised journal.

Reviewer #3

(Remarks to the Author)

The authors provide interesting experimental evidence on complex topological polar textures in marginally twisted hBN and TMDC homobilayers, with a focus on the interplay between moiré ferroelectricity and piezoelectricity. The highlight of the research is found in three polarity reversals observed across the domain walls for the out-of-plane polarization in twisted double trilayer hBN. On the other hand, in-plane polarization tends to align along the domain walls in vortex-like patterns. A similar effect is also observed in twisted TMDCs. I believe this work could be of interest to researchers focused on twisted 2D systems and ferroelectrics in general, but only after a major revision. Although this work presents novel insights into complex ferroelectric systems, a broader picture of the research and certain experimental information are missing, which would validate the reported results and emphasize the importance of the findings for other twisted 2D systems. Overall, the manuscript lacks the following points, which, if answered adequately, can make it suitable for publication in Nature Communications.

1. The authors provide vague information on material preparation, citing only one paper in which STM was used to determine the sample quality, which is not used in this work. The authors should provide more information on sample fabrication and material quality. Information such as whether the materials are synthetic or natural crystals, the assessment of the materials' defects, and the general interface quality after stacking? Particular interesting would be comment on defects, which could play important role in the ferroelectric behaviour of material.
2. Following information on material fabrication, how did the authors determine angles? As the only angle they are mentioning is less than 0.05% (which is a questionable number to be determined solely from optical image authors provided in SI). Did they use AFM images to determine angles, SHG or some other method? Also, information about the angle for WSe₂ and MoSe₂ is missing in the manuscript. Furthermore, the authors should provide AFM topographs to determine the thickness of the materials.
3. Also in terms of the thickness, authors should evaluate potential water layers formed on the surface, which usually interfere with the electric AFM modes, so authors should provide evidence of how they avoided that water formation or which process they used to minimize its effect?
4. The authors comment that the KPFM resolution is too small for 10-nm DW structures, but they did not specify their experimental resolution compared to standard PFM resolutions.
5. The authors did not use many references to describe how they connect OOP and IP with VPFM and LPFM, respectively, which makes the author's contribution to the method employed unclear. They should clarify more clearly which of the methods employed is their own advancement and what is existing state-of-the-art for the technique.
6. The authors' explanation for minimizing the buckling effect is described in detail; however, they should comment on how horizontal torsion is avoided in their calibration?
7. What is the range for small angles where the reported effects are visible? Authors should comment on what would change for slightly larger angles and, secondly, what is happening for other types of angles, such as H-type hBN? Did authors investigate larger angles (either experimentally or theoretically), because some broader picture should be given regarding findings (how reproducible they are for other types of samples).
8. As it seems that the paper is made only on a few samples with small statistics, authors should also reflect on the reproducibility of results; for example, did they confirm the results on other samples besides those reported in the manuscript? Also, as the author shows mainly figures size up to 1 μm , authors should provide information on how repetitive the results are within the samples they investigated, whether they see variations in some parameters (domain size, polarization amplitude or phase etc..) and what could be its origin.
9. Authors should comment on why is the domain size different in calculations from the experimental results for WSe₂ (Figure 3 and 4)?
10. The manuscript is highly focused on hBN results and generally lacks information on TMDCs results, as only a few paragraphs are dedicated to the result interpretation. Overall, the authors should provide a better summary of what exactly is the same between the materials, what is different, and why. For instance, in Figure 1, the signal from VPFM of hBN is stronger and, in principle, looks less noisy than that of MoSe₂ (Figure S10) and WSe₂ (Figure 3). Is there any specific reason for that?
11. Figure captions are not informative enough. Overall, the SI figures look rather confusing and lack better captions or paragraphs to describe them. For example, Figure S7 is comprised of almost 60 figures, which results in unreadable text. So authors should either divide on separate figures or invest better effort to clarify each figure.

Version 2:

Reviewer comments:

Reviewer #1

(Remarks to the Author)

I have read the referee reports, the answer from the authors, and the new version of the manuscript. The authors have

carefully responded to the queries from Referee 1 and Referee 3.

Referee 2 wrote that the article must have quantum mechanical, electronic band structure calculations: frankly, Referee 2 appears to want to review a different type of paper altogether. The present paper is on atomistic structure, and on its topological features, which are as important as electronic structure ones. This manuscript comprehensively build upon work by Bennett and others, which recently published in this same journal (Nature Communications 14, 1629 (2023)). In that sense, this work shows that Bennett's work ought to be modified to have the vector fields suddenly change at domain walls.

May I add that well-known groups such as Falko's (who also works on electronic structure) are working on this topic as well, and that the main experimental works on polar behavior on moire bilayers published in Nature Communications, too. Therefore, this journal indeed is the platform to continue this scientific discussion.

Given the detail of response to the questions aimed at clarifying what this manuscript is about, I recommend its publication without further delay.

Reviewer #2

(Remarks to the Author)

I reviewed the revised manuscript also in the light of all reviewers' comment. I believe that comments of Reviewer #2 have not been addressed. The use of ab initio simulations, either performed within DFT, post-HF or other quantum mechanical frameworks, is mandatory to elucidate the microscopic origin of the physical phenomena described in the manuscript. In the rebuttal letter, authors say that "our approach strategically employs quantum mechanical calculations where most impactful", meaning that they use some parameterisation derived from DFT calculations performed outside the present work. However, the theoretical framework used in the present work does not provide any quantum mechanical information of the processes underlying the observed polarisation. I do not understand why authors reply that "a full quantum-mechanical treatment in this context would not only be impractical but also offer little advantage over our current, community-accepted approach.". Full quantum treatment is indeed viable as suitable choices of the unit cell would limit the computational cost, as widely published in the literature; moreover, it would offer the advantage of, again, providing a quantum mechanical insight of the phenomenon, which is missing in the present version of the work. For this reason, I believe that the work does not meet the standards for publications in the journal. Without such insights, the work appears to provide a description just tailored on their system, which lacks quantum information on the mechanism of the charge distribution. Such information would provide experimental guidelines to expand the present study to other Moire systems, making the results useful for a more general audience, which is the audience of the Nature Communication journal.

Therefore, I cannot recommend the publication of the manuscript in the Nature Communication journal. I suggest authors to submit the manuscript to a more specialised journal.

Reviewer #3

(Remarks to the Author)

The authors have made adequate amendments to the manuscript, and I found it ready for publication in Nature Communications.

Unusual topological polar texture in moiré ferroelectrics

Response to Reviewers' Comments

We thank the reviewers for their well-informed comments. Below are our point-to-point responses to the comments. We have also made **several modifications to the manuscript, which are highlighted in red in the revised manuscript**. A list of major changes is summarized at the end of the report.

Reviewer #1

General Comment

This is a beautiful article combining experiment and theory on the polar states of marginally rotated bilayers. The noteworthy result is the discussion of piezoelectric effects, which are added to the ongoing discussion of topology provided by Bennett and coworkers (reference 36 on the manuscript); besides, while Reference 36 (a computational result) argues for a dipole moment slowly varying across Ab and BA domains, this work demonstrates that the topological polar texture is confined to joining sections of domain walls. It is also worth noting that the authors cross-check their assertions on three different homobilayers. The work will be of significance to the community working on topological properties of moiré materials.

Response

We sincerely thank the reviewer for recognizing the significance of our work.

1. Perhaps one omission from the authors is the work of Rebeca Engelke and coworkers (Phys. Rev. B 107, 125413 (2023)), where the topology of the domain walls is firmly established by experiment and theory as well. That team also has a variational model to describe distortions (Journal of Elasticity 154, 443 (2023)). I wonder if the authors can link the shading observed on their figure 2a (second row of results) to the color scheme shown by Engelke and coworkers in the PRB paper mentioned above.

Response

We thank the reviewer for pointing us to the nice works by Kim, Luskin et al. The shadings in Fig. 2a-c of our work are certain projections of the LPFM signal (phase and amplitude) near the DWs for different sample-cantilever angles. By combining them, we can reconstruct the full image of IP polarization field, as schematically illustrated by the black arrows in the top panel of Fig. 2a. Nevertheless, we agree a color map as in the suggested works would give a more direct view of the full IP polarization / Burgers vector field, which is now shown in Fig. R1 and Fig. S7 for the twisted hBN of the main text.

Figure R1. Reconstructed in-plane polarization of thBN1 with data from Fig.2a-c.

In the reference, we have added the two papers, and in the main text, we have now updated the related statements:

“Since the LPFM detects the component of the IP polarization perpendicular to the cantilever³⁰, the sine fit implies that the IP polarization near the DWs are along the DWs, thus forming clockwise and anti-clockwise vortex-like patterns (see the black arrows in the top panel of Fig. 2a and the color map of Fig. S7 for the reconstructed full IP polarization field), similar to the results obtained in graphene and TMDC moiré patterns³¹⁻³⁵.”

2. The work does support the conclusions and claims; the authors do a careful job in describing the experimental technique, which integrates out-of-plane and in-plane polar capabilities. No flaws are identified. The battery of results, including a symmetry analysis, do indicate clever ways to separate piezo responses from ferroelectric ones. Perhaps an outstanding question is whether the force microscope tip contributes to the observed piezoelectric strain.

Response

We thank the reviewer for raising this critical question. Indeed, since PFM operates in contact mode, tip-sample forces are inherently unavoidable. And a significant force applied by the tip can influence the observed piezoelectric strain, either by introducing additional deformation or partially relaxing the built-in strain of the moiré superlattice. Notably, prior studies have demonstrated that AFM tip-induced mechanical forces can effectively modify interlayer stacking and moiré patterns in two-dimensional materials (Nat. Nanotech. 2018, 13, 204-208, Nat. Mater. 2022, 21, 621-626, Nano Lett. 2025, 25, 1584-1592).

Throughout our experiment, we intentionally applied a small force (<10 nN) onto the samples to minimize extrinsic effects brought by the tip. We have also conducted force dependence PFM measurements on a 0.035° twisted hBN sample. The overall moiré structures and line profiles (Fig. R2 and Fig. S20) look almost independent of the force in the 5.9-23.5 nN regime.

In the method, we now specify the tip force that we applied in the experiment.

“Throughout the PFM measurements, the tip-sample contact force was maintained below 10 nN to minimize mechanical perturbation of the sample.”

Figure R2. Force-dependent VPFM measurements of thBN2. a-i, VPFM phase images, amplitude images and corresponding linecuts with 5.9 nN (a-c), 11.8 nN (d-f) and 23.5 nN (g-i) normal force, respectively. No significant change is observed for different forces in this range.

3. Having said the above, the article can be followed very clearly, and it is my impression that its descriptions are very detailed so as to be reproducible.

Response

We thank the reviewer again for their positive feedback.

Reviewer #2

General Comment

Authors describe an experimental study polar texture in moiré ferroelectrics whereas some simulations are provided with the aim to complement the experimental data. The topic looks properly inserted into the existing literature. The experimental setup seems properly detailed. The results looks properly presented and discussed. The results provide new physical insights into the electronic band topology in twisted transition metal dichalcogenides (TMDs).

Response

We appreciate the reviewer's acknowledgement that our work was properly presented and discussed. Our results provide the first experimental evidence of complex polar textures in moiré ferroelectrics, which may offer new insights into the electronic band topology in twisted TMDs.

1. However, the electronic properties of moiré TMDs are largely studied and many works focus on specific aspects which are suitable for more specialistic journals, as it is the case of the present manuscript.

Response

We thank the reviewer's comment on the electronic properties of moiré TMDCs. While it is true that the electronic properties of moiré TMDCs with twist angles of 1-5°, particularly their band structure, topology features and correlated phenomena have been extensively investigated, the marginally twisted regime ($<1^\circ$) remains significantly less explored. This gap persists partially due to the computational challenges posed by lattice reconstruction in this regime.

More importantly, **our work represents a distinct research direction: the experimental investigation of the topological polar structure in moiré ferroelectrics with small twist angles.** This emerging field, separate from conventional electronic property studies of moiré TMDCs, has recently gained considerable attention, particularly in the field of ferroelectricity. Examples include theoretical prediction of polar meron-antimeron networks in twisted hBN (Ref. 38), vortices in large twist-angle MoS₂ (Ref. 18), topological phases in polar oxide nanostructures (Ref. 3) etc. In this study, we reveal alternating out-of-plane polarization in the domain regions and vortex-like in-plane polarization patterns on the domain wall boundaries, indicating the formation of meron and antimeron networks in moiré ferroelectrics. Moreover, by suppressing the cantilever buckling effect or applying antisymmetrization to the VPFM images, we observe multiple out-of-plane polarization reversals across the domain walls, which indicates the competition between moiré ferroelectricity and piezoelectricity.

These polar textures are also significant to the electronic properties of twisted TMDCs, which are highly sensitive to the moiré potential landscape. Therefore, our findings not only provide the first experimental evidence of complex polar textures in moiré ferroelectrics but may also offer new insights into the electronic band topology in twisted TMDCs.

Accordingly, we have modified the abstract's concluding sentence to more precisely reflect this connection. "These results provide the first experimental evidence of complex polar textures in moiré ferroelectrics, **which may offer** new insights into the electronic band topology in twisted transition metal dichalcogenides (TMDCs)."

2. Finally, the lack of quantum mechanical calculations makes the manuscript unsuitable for publication in the Nature Communication journal. In fact, quantum mechanical calculations are necessary to provide a complete insight into the physical phenomena described in the manuscript; in fact, such kind of calculations are commonly employed in scientific articles regarding electronic properties of moiré van der Waals heterostructures.

Response

We appreciate the reviewer's comment regarding the calculations. Our theoretical modeling proceeds in two steps. First, we determine the atomic structure by elastic theory based on parameters derived from DFT calculations. Second, we compute interlayer charge transfer and piezoelectric polarization from that atomic structure.

In the first step, elastic theory is widely adopted for modeling the atomic structure of moiré superlattices. The original article (Phys. Rev. B 98, 224102) has been cited more than 300 times. This model reproduces all essential features of moiré superlattice structure, including domain and domain-wall formation. In the small-twist-angle limit ($\theta \ll 1^\circ$) relevant to our study, domain and domain-wall formation effectively reduce to a one-dimensional problem that can be solved almost analytically. All parameters in this model are obtained from DFT calculations, a quantum-mechanical method. Notably, a full quantum-mechanical relaxation of such large moiré superlattice is computationally prohibitive due to the large number of atoms involved. Our method, which captures the essential characteristics of the moiré structure, is thus well-suited to analyze the experimental results.

In the second step, the interlayer charge transfer (ferroelectric polarization) is also determined by density functional theory calculations with modern theory of polarization for each stacking configuration. We agree, in principle, a full quantum-mechanical calculation of the electronic structure of the large relaxed

moiré superlattice could give a complete insight into the phenomena we observed. However, it is beyond current state-of-the-art computational capability. Even if these calculations were feasible, they would not elucidate the underlying origin of the observed polarization. In contrast, our approach, validated in Ref. 40, provides essential mechanistic insight into the experimentally measured polarization.

In summary, while our approach strategically employs quantum mechanical calculations where most impactful, a full quantum-mechanical treatment in this context would not only be impractical but also offer little advantage over our current, community-accepted approach.

Therefore, I cannot recommend the publication of the manuscript in the Nature Communication journal. I suggest authors to submit the manuscript to a more specialised journal.

Response

We appreciate the reviewer's time and careful evaluation of our manuscript. While we respect their perspective, we believe our study aligns well with Nature Communications' scope by presenting fundamental advances in moiré ferroelectrics that bridge materials physics, nanoscale characterization, and device-relevant phenomena. Our discovery of topological polar textures offers new insights with potential implications across these interdisciplinary areas.

We have carefully addressed all technical concerns through our detailed responses, demonstrating both the rationality of our methodology and the novelty of our findings. We would be most grateful if the reviewer could reconsider their recommendation in light of these clarifications and the novel contributions of our work to the field of moiré ferroelectrics.

Reviewer #3

General Comment

The authors provide interesting experimental evidence on complex topological polar textures in marginally twisted hBN and TMDC homobilayers, with a focus on the interplay between moiré ferroelectricity and piezoelectricity. The highlight of the research is found in three polarity reversals observed across the domain walls for the out-of-plane polarization in twisted double trilayer hBN. On the other hand, in-plane polarization tends to align along the domain walls in vortex-like patterns. A similar effect is also observed in twisted TMDCs. I believe this work could be of interest to researchers focused on twisted 2D systems and ferroelectrics in general, but only after a major revision. Although this work presents novel insights into complex ferroelectric systems, a broader picture of the research and certain experimental information are missing, which would validate the reported results and emphasize the importance of the findings for other twisted 2D systems. Overall, the manuscript lacks the following points, which, if answered adequately, can make it suitable for publication in Nature Communications.

Response

We thank the reviewer for the careful reading and very helpful suggestions.

1. The authors provide vague information on material preparation, citing only one paper in which STM was used to determine the sample quality, which is not used in this work. The authors should provide more information on sample fabrication and material quality. Information such as whether the materials are synthetic or natural crystals, the assessment of the materials' defects, and the general interface quality after stacking? Particular interesting would be comment on defects, which could play important role in the ferroelectric behaviour of material.

Response

We thank the reviewer for their valuable suggestions regarding material information and characterization.

Below are the detailed clarifications:

(1) Material sources and quality:

- High quality highly oriented pyrolytic graphite (HOPG), WSe₂ and MoSe₂ crystals are sourced from HQ Graphene (synthetic, high-purity).
- hBN crystals were grown by the coauthors (Prof. Taniguchi and Prof. Watanabe) using established high-pressure methods, ensuring low defect densities.

(2) Interface quality assessment:

- After stacking, we perform large-scale LPFM scans to identify regions with clean interfaces (from the topography) and desired twist angles.
- Selected areas are further refined via AFM tip cleaning (contact mode) to minimize surface contaminants before detailed KPFM and vector PFM studies.

(3) Role of defects:

Defects, such as point defects, grain boundaries, wrinkles etc., in general play an important role in the ferroelectric behavior by modulating the free energy profile. In the case of 2D materials and their moiré superlattices, a particular type of defect, i.e. interlayer dislocations are commonly observed (e.g. PNAS 2013, 110, 11256-11260, Nat. Photon. 2022, 16, 469-474). These defects are faintly visible in topography, but can have a huge impact on the electronic, optical and mechanical properties of the system. For instance, in twisted multilayer graphene moiré superlattices, we have demonstrated (Nano Lett. 2025, 25, 1584-1592) that they can interact with moiré domain walls and effectively tune the polar vortices and electronic properties. Similar defects may also present in twisted hBN and TMDCs, perturbing the polar textures and electronic properties. Future studies correlating atomic-scale defect mapping with piezoresponse measurements could enable defect engineering to tailor moiré ferroelectrics for device applications.

Accordingly, in the Methods, we have added information of the crystals and additional treatments to ensure clean interfaces. “High quality highly oriented pyrolytic graphite (HOPG), WSe₂ and MoSe₂ crystals are sourced from HQ Graphene. Atomically thin flakes were obtained by mechanical exfoliation, with thicknesses initially identified through optical contrasts and subsequently confirmed by AFM. All measured devices (including 5 twisted hBN (thBN1-5), 2 twisted WSe₂ and 2 twisted MoSe₂ devices) were fabricated using a modified ‘tear-and-stack’ method⁴⁷. Detailed stacking procedures, typical optical images of twisted hBN and TMDCs are presented in Fig. S1&S2. Following stacking, we performed large-scale LPFM scans to identify regions with clean interfaces (from the topography) and desired twist angles. Twist angles (θ) were calculated from the measured moiré periodicity (a_M) in the LPFM images, following the relation $a_M = a/(2 \sin(\theta/2))$, where a is the lattice constant of the constituent 2D material. For moiré patterns exhibiting clearly non-equilateral triangular morphologies, we adopted the same angle determination strategy as described in Ref.⁴⁸. Selected areas are further refined via AFM tip cleaning to minimize surface contaminants before detailed KPFM and vector PFM studies.”

In the Supplementary, we have added schematics of detailed stacking procedures (Fig. S1).

In the Discussion, we have added a short discussion of the effects of defects. “Beyond twist angle and material-specific properties, the polar textures in moiré superlattices may also be influenced by intrinsic and extrinsic defects. For instance, point defects and grain boundaries can act as pinning sites for electric polarizations^{41,42}, creating localized dipole fluctuations and altering the switching dynamics. Another example is interlayer dislocations in 2D materials and their heterostructures, often faintly visible in topography, may alter the strain fields and electrostatic landscapes of the moiré pattern⁴³, thereby modifying the polar textures and electronic properties.”

2. Following information on material fabrication, how did the authors determine angles? As the only angle they are mentioning is less than 0.05% (which is a questionable number to be determined solely from optical image authors provided in SI). Did they use AFM images to determine angles, SHG or some other method? Also, information about the angle for WSe2 and MoSe2 is missing in the manuscript. Furthermore, the authors should provide AFM topographs to determine the thickness of the materials.

Response

We appreciate the reviewer’s inquiry regarding twist angle determination and materials’ thicknesses. The twist angles in our samples were calculated using the moiré periodicity (a_M , Fig. 1b) observed in PFM images, following the relation $a_M = a/(2 \sin(\theta/2))$, where a is the lattice constant of the constituent 2D material and θ is the twist angle. For example, in twisted hBN ($a = 0.25$ nm), a moiré periodicity of 287 nm would correspond to a twist angle of 0.05°. For moiré patterns exhibiting clearly non-equilateral triangular morphologies, we adopted the same angle determination strategy as described in Ref.⁴⁹.

We have now specified the calculated twist angles in figure captions and upon first mention in the main text. Exemplary topographies verifying thicknesses of hBN, WSe₂ and MoSe₂ are now included in Supplementary Fig. S2.

We have also added the following sentences about the determination of twist angles in the Methods and the caption of Fig. 1b:

“Twist angles (θ) were calculated from the measured moiré periodicity (a_M) in the LPFM images, following the relation $a_M = a/(2 \sin(\theta/2))$, where a is the lattice constant of the constituent 2D material. For moiré patterns exhibiting non-equilateral triangular morphologies, we adopted the same angle determination strategy as described in Ref.⁴⁸.”

“The relation between twist angles (θ) and measured moiré periodicity (a_M) of twisted hBN moiré superlattice is given by $a_M = a/(2 \sin(\theta/2))$, where a is the lattice constant of hBN.”

3. Also in terms of the thickness, authors should evaluate potential water layers formed on the surface, which usually interfere with the electric AFM modes, so authors should provide evidence of how they avoided that water formation or which process they used to minimize its effect?

Response

This is a very insightful question. Water layers formed between the tip and the sample surface at ambient conditions are believed to play a crucial role in SPM measurements. In terms of contact-mode vector PFM imaging, water layers can modulate PFM signals through multiple mechanisms, including capillary force interactions, dielectric screening, electric field redistribution, and electrochemical processes. As noted, the literature presents some inconsistency in reported effect of humidity on such measurements, possibly due to sample variations and other measurement conditions.

In our experiments, we maintained a moderately low relative humidity (RH, ~35%) using a humidity control system. While the primary focus of this work is the polar structure within the moiré superlattices, where humidity effects were observed not to alter our key conclusions, we acknowledge the potential influence of water layers on quantitative electromechanical characterizations. Supporting this, Fig. R3 (Fig. S18) demonstrates the robustness of moiré patterns and sign reversals of VPFM near domain walls in a 0.035° twisted double trilayer hBN sample (thBN2) across a humidity range (19% - 68% RH). Intriguingly, we observed enhanced spatial resolution at higher relative humidity, suggesting a humidity-dependent tip-sample interaction mechanism. We also perform KPFM measurements on a different region of thBN2 (the twist angle of this region is around 0.02°) at different humidities (20% - 70% RH), no significant difference in the potential difference between adjacent domains (AB, BA) is observed (Fig. R4 and Fig. S19).

Lastly, for quantitative piezoresponse measurements (e.g. precise determination of piezoelectric coefficients, polarization switching field etc.), more stringent environmental controls, such as inert gas purging or high vacuum would indeed be necessary, which could be a future research direction.

In the Methods, we now specify the environmental conditions of our experiment. “Three AFM modes were adopted in this work, i.e., VPFM, LPFM, and KPFM, all performed on an Oxford Instruments Asylum Research MFP-3D Origin AFM **at ambient conditions with a relative humidity of ~35%.**”

“More VPFM and KPFM results of twisted hBN (thBN2) under different relative humidities can be found in Figs. S18&S19. Figure S20 shows similar VPFM results of twisted hBN under different applied normal forces.”

Figure R3. VPFM measurements of thBN2 under different relative humidity (19%, 35% and 68%). a-i, VPFM phase images, amplitude images and corresponding linecuts under 19% (a-c), 35% (d-f) and 68% (g-i) RH, respectively. The spatial resolution increases with relative humidity in this range. The twist angle of this region is around 0.035° .

Figure R4. KPFM measurements of thBN2 under different relative humidity (20%, 37%, 58% and 70%). a-d, Surface potential image and corresponding line cut with 20%, 37%, 58% and 70% RH. The potential difference between AB and BA stackings keeps almost constant (~ 105 mV). Note the KPFM is performed on a different region of thBN2, where the twist angle is around 0.02° .

4. The authors comment that the KPFM resolution is too small for 10-nm DW structures, but they did not specify their experimental resolution compared to standard PFM resolutions.

Response

Thanks for pointing this out. In response, we have now specified the experimental resolutions of KPFM and PFM in the revised manuscript. As shown in Fig. R5 and Fig. S3, while PFM achieves sufficient resolution to resolve the 10-nm domain wall structures, KPFM under our experimental conditions is indeed limited for these fine features.

Figure R5. In-situ VPFM (a, b) and KPFM (c) measurements on thBN2. The corresponding linecuts demonstrate spatial resolutions of 9 nm (VPFM) and 50 nm (KPFM), determined from the transition across the same sharp topographic step. The twist angle of this region is around 0.065° .

We have added Fig. S3 to the supplementary and updated the related statements in the revised manuscript.

“To reconstruct the full polarization field with higher resolution, we switch to vector PFM for the rest of this study (see Fig. S3 for experimental comparison of the distinct spatial resolutions achieved by in-situ KPFM and PFM measurements on thBN2).”

5. The authors did not use many references to describe how they connect OOP and IP with VPFM and LPFM, respectively, which makes the author's contribution to the method employed unclear. They should clarify more clearly which of the methods employed is their own advancement and what is existing state-of-the-art for the technique.

Response

We thank the reviewer for raising this critical question. We acknowledge that the conventional assumption of direct correspondence between VPFM/LPFM signals and OOP/IP polarizations (without cross-talk) is commonly adopted in the literature (e.g., Ref. 9, ACS Nano 2018, 12, 4976-4983). However, this approach becomes problematic when studying systems like ours where the OOP polarization is rather weak. In this case, the cantilever buckling effects (IP polarization contributions to VPFM) could be significant and hard to avoid in normal PFM.

Indeed, previous studies have developed several approaches to address this challenge. This include using stiffer cantilever to minimized buckling of the cantilever, carefully aligning the cantilever perpendicular to the IP polarization direction, optimizing the laser position, and angle-dependent measurement protocols that mathematically decouple the competing contributions (e.g. Ref. ^{36,37}).

In our investigation of twisted hBN or TMDCs moiré superlattices, we have developed two system-specific innovations:

- (1) DW-alignment method: Our angle-dependent LPFM measurements revealed that the IP polarization consistently aligns with the DW directions. This unique property allows us to effectively suppress buckling effects by simply aligning the cantilever perpendicular to the DWs, as demonstrated in Fig. 1d&e.
- (2) Symmetry-based analysis: The antisymmetric distribution of OOP polarization versus the symmetric distribution of IP polarization with respect to the SP stacking in our system enable extraction of intrinsic OOP polarization image through antisymmetrization of the VPFM images.

To clarify these points, we have added the following to the revised manuscript.

“In contrast to existing approaches for suppressing cantilever buckling effects^{36,37}, the alignment of IP polarization along DW directions in our system enables straightforward suppression by orienting the cantilever perpendicular to the DWs, as for the highlighted DW (DW1) in Fig. 1d&e.”

“Alternatively, the intrinsic OOP polarization can be extrapolated from symmetry analysis in this particular system”.

6. The authors' explanation for minimizing the buckling effect is described in detail; however, they should comment on how horizontal torsion is avoided in their calibration?

Response

We thank the reviewer for raising this important technical point. In contact-mode PFM measurements, an AC excitation is applied to the tip, the in-plane polarization perpendicular to the cantilever causes torsional oscillations at this particular frequency, resulting in the signal detected in LPFM (Fig. S9). This torsion can also give rise to undesired VPFM signals (cross-talk) if the photodiode is rotated slightly with respect to the plane of the deflected beam (Rev. Sci. Instrum. 2007, 78, 016101). We calibrated our PFM system with a standard ferroelectric sample (periodically poled lithium niobate, PPLN) to ensure this crosstalk is negligible. Moreover, this torsion induced VPFM signal would also be symmetric with respect to SP stackings, which conflicts with our observations.

Besides, horizontal torsion can also occur due to lateral friction between the tip and sample, particularly when scanning rough surfaces or applying high normal forces. This can indeed introduce artifacts to the LPFM signals. In our work, the twisted hBN/TMDC samples have atomically smooth terraces as confirmed by the topography, and we maintained a low normal force (<10 nN) to reduce lateral friction forces. Moreover, we check the reproducibility of forward and backward scans, they usually overlap with each other, indicating negligible friction force effect to our PFM measurements.

7. What is the range for small angles where the reported effects are visible? Authors should comment on what would change for slightly larger angles and, secondly, what is happening for other types of angles, such as H-type hBN? Did authors investigate larger angles (either experimentally or theoretically), because some broader picture should be given regarding findings (how reproducible they are for other types of samples).

Response

Thanks for the very insightful suggestion. We have now performed similar measurements on twisted hBN samples with slightly larger twist angles up to around 0.375° , corresponding to moiré periodicities of 38.2 nm. The cantilever buckling effect is relatively weak during the measurements. As shown in Fig. R6 and Fig. S16, while the imaging quality decreases with smaller moiré periodicities, we find the polar textures remain qualitatively similar (maintaining meron-like characteristics and multiple sign reversals), though with less confined domain walls at larger angles.

Regarding H-type twisted hBN, our experiments reveal a clear distinction from R-type configurations. We have fabricated several marginally twisted hBN samples with monolayer hBN terraces at the twisted interface. While we observe regular moiré patterns and PFM contrast in R-type regions, no detectable PFM signal is found in H-type regions (Fig. R7). This striking contrast suggests either an unavoidable or unintentional relaxation to uniform centrosymmetric AA' stacking in the H-type stacked regions. In fact, there is an ongoing debate regarding moiré formation in H-stacked systems (Nat. Comm. 2021, 12, 347). Further investigations are demanded for a better understanding.

Figure R6. VPFM measurements of thBN02 with nonuniform twist angles. **a**, VPFM phase image of large scan. **b-e**, Zoom-in scans of the selected regions with twist angles vary from 0.064° to 0.375° . The observed multiple sign reversal is robust over the twist angle range of 0.064° to 0.375° . VPFM scans on this sample also show a weak cantilever buckling effect as Fig. S13.

Figure R7. In-situ KPFM and VPFM measurements of a marginally twisted hBN with a monolayer terrace at the twisted interface. Topography (**a**), VPFM phase (**b**), VPFM phase (**c**) and surface potential (**d**) images and corresponding line cuts of twisted hBN. Line cuts of topography, VPFM phase, VPFM amplitude and surface potential reveal a clear distinction between the R-type and H-type configurations. Twist angles vary from 0.016° to 0.031° .

In the discussion, we have now added a short paragraph as below:

“Figures S15-17 present PFM results of a nonuniformly twisted hBN sample (thBN2&thBN5). While the twist angle varies from 0.014° to 0.375° , the meron-like polarization distribution and multiple OOP polarization reversals persist.”

8. As it seems that the paper is made only on a few samples with small statistics, authors should also reflect on the reproducibility of results; for example, did they confirm the results on other samples besides those reported in the manuscript? Also, as the author shows mainly figures size up to $1\ \mu\text{m}$, authors should provide information on how repetitive the results are within the samples they investigated, whether they see variations in some parameters (domain size, polarization amplitude or phase etc.) and what could be its origin.

Response

Yes, for each material system, we have reproduced the results on several samples, including 3 twisted hBN samples, 2 twisted WSe₂ samples and 2 twisted MoSe₂ samples. Each sample can be as large as $50 \times 50\ \mu\text{m}$ (bilayer region), and there may be several isolated regions forming moiré superlattices with moiré periodicity larger than $1\ \mu\text{m}$. Across all samples, we consistently observe meron-like polar textures and multiple sign reversals near DWs in regions exhibiting moiré patterns with periodicities larger than $50\ \text{nm}$.

For those with moiré periodicities larger than $200\ \text{nm}$, the ferroelectric and piezoelectric contributions to the PFM signals remain consistent, independent of the moiré size. This observation aligns with the saturation of domain wall width in this regime, as shown in Fig. R8. However, as the twist angle increases beyond this range, we note significant changes in the DW ratio, accompanied by a corresponding decrease in the PFM signal amplitude (Fig. R6), reflecting reduced polarizations. These variations appear to originate from differences in lattice reconstruction behavior across the angular range studied.

Figure R8. VPFM measurement of a twisted double trilayer hBN sample with nonuniform twist angles. **a,b**, VPFM phase and amplitude images of twisted hBN. **c**, Line cuts of amplitude across nearly vertical AB/BA domain walls in **b**. The amplitude

signals are offset for clarity. No significant changes are observed as the moiré periodicity (a_M) increases from 269 nm ($\theta = 0.053^\circ$) to 1058 nm ($\theta = 0.014^\circ$).

We have now added information about the number of devices investigated in the Methods.

“All measured devices (including 5 twisted hBN (thBN1-5), 2 twisted WSe₂ and 2 twisted MoSe₂ devices) were fabricated using a modified ‘tear-and-stack’ method⁴⁷.”

Discussions of the twist angle (moiré size) dependence, effects on the polarization phase and amplitude are now included in Fig. S15.

9. Authors should comment on why is the domain size different in calculations from the experimental results for WSe₂ (Figure 3 and 4)?

Response

Thanks for the suggestion. We have now performed new calculations using the same twist angle (0.06°) as in our experimental measurements (Fig. 3). These revised results, presented in the updated Figure 4, quantitatively agree with the experimental results by preserving all the key physical phenomena we originally reported. Importantly, this consistency further validates that the observed phenomena is robust across a wide range of twist angles ($0.01^\circ - 0.375^\circ$).

10. The manuscript is highly focused on hBN results and generally lacks information on TMDCs results, as only a few paragraphs are dedicated to the result interpretation. Overall, the authors should provide a better summary of what exactly is the same between the materials, what is different, and why. For instance, in Figure 1, the signal from VPFM of hBN is stronger and, in principle, looks less noisy than that of MoSe₂ (Figure S10) and WSe₂ (Figure 3). Is there any specific reason for that?

Response

We thank the reviewer for raising this question. In the manuscript, we provide a comparison of the VPFM signals between twisted WSe₂ homobilayer and twisted double trilayer hBN, where we stated “Clearly, the AB and BA stacking domains also exhibit the expected 180° phase contrast, with VPFM amplitudes of ~ 7 pm, nearly one third of that observed in twisted hBN. Near the DWs, the features are less pronounced but still distinguishable”. This suggests that both ferroelectric and piezoelectric polarizations of twisted WSe₂ are smaller than those of twisted hBN. The reduced ferroelectric polarization can be understood as the reduced orbital overlaps (charge transfers) of twisted WSe₂. The suppressing of piezoelectric polarization is due to the reduced piezoelectric effect as a result of the smaller piezoelectric coefficient.

Regarding the noise floors, the measured noise levels of the presented twisted WSe₂ (Fig. 3) and MoSe₂ (Fig. S14) are comparable to those in twisted hBN (Fig. 1, see enlarged view in Fig. R9). The apparently lower noise in twisted hBN arises from its significantly enhanced piezoelectric response, which yields a superior signal-to-noise ratio rather than reduced intrinsic noise. We acknowledge that the differences in sample fabrication variability, AFM tip conditions, and other environmental conditions can affect the

noise of our measurements. For example, WSe₂ and MoSe₂ are chemically less stable than hBN, which might decrease the signal quality through surface degradation effects.

Figure R9. Comparison of VPFM noise floors for twisted hBN (Fig. 1), WSe₂ (Fig. 3) and MoSe₂ (Fig. S14). The noise floors are roughly consistent across different materials (3-5 pm).

In the discussion, we have now added a short paragraph discussing the difference between hBN and TMDCs. “Although these universal features appear across different moiré systems, their specific manifestations are influenced by intrinsic material properties. Twisted hBN exhibits stronger interlayer charge transfer and piezoelectric response due to its ionic character, while twisted WSe₂ and MoSe₂ show more modest charge redistribution but stronger band structure modulation through orbital hybridizations. This difference leads to hBN’s more pronounced electrostatic potential variations versus TMDCs’ stronger coupling to electronic states.”

11. Figure captions are not informative enough. Overall, the SI figures look rather confusing and lack better captions or paragraphs to describe them. For example, Figure S7 is comprised of almost 60 figures, which results in unreadable text. So authors should either divide on separate figures or invest better effort to clarify each figure.

Response

We thank the reviewer for comments on the figures and figure captions. We have now added more information to the figure captions of the supplementary. We also delete part of the subfigures in Fig. S7, divided the remaining to two separate figures (Fig. S10&S11 in the revised supplementary). We hope it clarifies the reviewer’s concern.

List of major changes (main text changes highlighted in red):

- **Abstract:** we have modified the abstract's concluding sentence to more precisely reflect the connection between our findings and the electronic band topology. "These results provide the first experimental evidence of complex polar textures in moiré ferroelectrics, **which may offer** new insights into the electronic band topology in twisted transition metal dichalcogenides (TMDCs).
- **Main text:** We have updated relevant statements on spatial resolution comparison, reconstructed full IP polarization field and system-specific ways of suppressing cantilever buckling effects.
"To reconstruct the full polarization field with higher resolution, we switch to vector PFM for the rest of this study (see Fig. S3 for experimental comparison of the distinct spatial resolutions achieved by in-situ KPFM and PFM measurements on thBN2)."
"Since the LPFM detects the component of the IP polarization perpendicular to the cantilever³⁰, the sine fit implies that the IP polarization near the DWs are along the DWs, thus forming clockwise and anti-clockwise vortex-like patterns (see the black arrows in the top panel of Fig. 2a and the color map of Fig. S7 for the reconstructed full IP polarization field), similar to the results obtained in graphene and TMDC moiré patterns³¹⁻³⁵."
"In contrast to existing approaches for suppressing cantilever buckling effects^{36,37}, the alignment of IP polarization along DW directions in our system enables straightforward suppression by orienting the cantilever perpendicular to the DWs, as for the highlighted DW (DW1) in Fig. 1d&e."
- **Discussion:** We have added two paragraphs to discuss twist angle dependence of PFM results, materials-specific differences and effects of defects on the phenomena we observed.
"Figures S15-17 present PFM results of a nonuniformly twisted hBN sample (thBN2&thBN5). While the twist angle varies from 0.014° to 0.375°, the meron-like polarization distribution and multiple OOP polarization reversals persist."
"Although these universal features appear across different moiré systems, their specific manifestations are influenced by intrinsic material properties. hBN exhibits stronger interlayer charge transfer and piezoelectric response due to its ionic character, while WSe₂ and MoSe₂ show more modest charge redistribution but stronger band structure modulation through orbital hybridizations. This difference leads to hBN's more pronounced electrostatic potential variations versus TMDCs' stronger coupling to electronic states."
"Beyond twist angle and material-specific properties, the polar textures in moiré superlattices may also be influenced by intrinsic and extrinsic defects. For instance, point defects and grain boundaries can act as pinning sites for electric polarizations^{41,42}, creating localized dipole fluctuations and altering the switching dynamics. Another example is interlayer dislocations in 2D materials and their

heterostructures, often faintly visible in topography, may alter the strain fields and electrostatic landscapes of the moiré pattern⁴³, thereby modifying the polar textures and electronic properties.”

- **Methods:** We have added information of the crystals, twist angle determination strategies, additional interface cleaning procedures, and AFM measurement conditions.

“**Sample fabrication.** High quality highly oriented pyrolytic graphite (HOPG), WSe₂ and MoSe₂ crystals are sourced from HQ Graphene. Atomically thin flakes were obtained by mechanical exfoliation, with thicknesses initially identified through optical contrasts and subsequently confirmed by AFM. All devices were fabricated using a modified ‘tear-and-stack’ method⁴⁷. Detailed stacking procedures, typical optical images of twisted hBN and TMDC are presented in Fig. S1&S2. Following stacking, we performed large-scale LPFM scans to identify regions with clean interfaces (from the topography) and desired twist angles. Twist angles (θ) were calculated from the measured moiré periodicity (a_M) in the LPFM images, following the relation $a_M = a/(2 \sin(\theta/2))$, where a is the lattice constant of the constituent 2D material. For moiré patterns exhibiting non-equilateral triangular morphologies, we adopted the same angle determination strategy as described in Ref.⁴⁸. Selected areas are further refined via AFM tip cleaning to minimize surface contaminants before detailed KPFM and vector PFM studies.”

“**Atomic force microscopy measurements.** Three AFM modes were adopted in this work, i.e., VPFM, LPFM, and KPFM, all performed on an Oxford Instruments Asylum Research MFP-3D Origin AFM at ambient conditions with a relative humidity of ~35%. ASYELEC-01-R2 probes coated with 5-nm Ti and 20-nm Ir were used in all modes with a spring constant of 2.8 N/m. Typical free resonance frequency is ~75 kHz. VPFM and LPFM contact resonance frequency are ~300 kHz and ~780 kHz, respectively. Throughout the PFM measurements, the tip-sample contact force was maintained below 10 nN to minimize mechanical perturbation of the sample. In KPFM measurement, the dual pass amplitude modulation is employed to capture the topography and surface potential difference between tip and sample, respectively. More VPFM and KPFM results of twisted hBN (thBN2) under different relative humidities can be found in Figs. S18&S19. Figure S20 shows similar VPFM results of twisted hBN under different applied normal forces. To minimize the potential shift, a zero-order flattening process is applied to the raw surface potential data.”

- **References:** We have added Refs. 34,35,41,42,43,48 and removed Ref. 39 (previous version) to the revised manuscript.

- **Figures:** We have added the following sentences about the determination of twist angles in the caption of Fig. 1b. “The relation between twist angles (θ) and measured moiré periodicity (a_M) of twisted hBN moiré superlattice is given by $a_M = a/(2 \sin(\theta/2))$, where a is the lattice constant of hBN.” We have updated the simulation results in Fig. 4 using the same twist angle (0.06°) as in our experimental measurements (Fig. 3).
- **Supplementary:** We have added several supporting figures (Figs. S1, S3, S7 and S15-S20) to the revised supplementary. We have also added additional information to the figure captions, deleted part of the subfigures in Fig. S7, divided the remaining into two separate figures (Fig. S10&S11).

Unusual topological polar texture in moiré ferroelectrics

Response to Reviewers' Comments

We thank the reviewers for their well-informed comments. Below are our point-to-point responses to the comments.

Reviewer #1

General Comment

I have read the referee reports, the answer from the authors, and the new version of the manuscript. The authors have carefully responded to the queries from Referee 1 and Referee 3.

Referee 2 wrote that the article must have quantum mechanical, electronic band structure calculations: frankly, Referee 2 appears to want to review a different type of paper altogether. The present paper is on atomistic structure, and on its topological features, which are as important as electronic structure ones. This manuscript comprehensively build upon work by Bennett and others, which recently published in this same journal (Nature Communications 14, 1629 (2023)). In that sense, this work shows that Bennett's work ought to be modified to have the vector fields suddenly change at domain walls.

May I add that well-known groups such as Falko's (who also works on electronic structure) are working on this topic as well, and that the main experimental works on polar behavior on moire bilayers published in Nature Communications, too. Therefore, this journal indeed is the platform to continue this scientific discussion.

Given the detail of response to the questions aimed at clarifying what this manuscript is about, I recommend its publication without further delay.

Response

We sincerely thank the reviewer for recognizing and recommending our work.

Reviewer #2

General Comment

I reviewed the revised manuscript also in the light of all reviewers' comment. I believe that comments of Reviewer #2 have not been addressed. The use of ab initio simulations, either performed within DFT, post-HF or other quantum mechanical frameworks, is mandatory to elucidate the microscopic origin of the physical phenomena described in the manuscript. In the rebuttal letter, authors say that "our approach strategically employs quantum mechanical calculations where most impactful", meaning that they use some parameterisation derived from DFT calculations performed outside the present work. However, the theoretical framework used in the present work does not provide any quantum mechanical information of the processes underlying the observed polarisation. I do not understand why authors reply that "a full quantum-mechanical treatment in this context would not only be impractical but also offer little advantage over our current, community-accepted approach.". Full quantum treatment is indeed viable as suitable choices of the unit cell would limit the computational cost, as widely published in the literature; moreover, it would offer the advantage of, again, providing a quantum mechanical insight of the phenomenon, which is missing in the present version of the work. For this reason, I believe that the work does not meet the standards for publications in the journal. Without such insights, the work appears to provide a description just tailored on their system, which lacks quantum information on the mechanism of the charge distribution. Such information would provide experimental guidelines to expand the present study to other Moire systems, making the results useful for a more general audience, which is the audience of the Nature Communication journal.

Therefore, I cannot recommend the publication of the manuscript in the Nature Communication journal. I suggest authors to submit the manuscript to a more specialised journal.

Response

We sincerely appreciate the reviewer's time and thoughtful feedback. While we understand the importance of full quantum mechanical insights in elucidating the microscopic origins of the observed phenomena, we respectfully disagree with the feasibility of performing full ab initio calculations for the system under study (**marginally** twisted moiré superlattices).

As we mentioned in our last response letter, the primary challenge of a full quantum treatment lies in the enormous computational cost associated with the system size (a single **0.06° twisted bilayer WSe₂ moiré unit cell contains over 5 million atoms**), making such calculations intractable with current methods as far as we know. Developing more advanced computational techniques (e.g. machine learning-assisted approaches) to address this challenge remains an active area of research. Furthermore, while a full quantum mechanical treatment would directly yield the total electric polarization, our semiclassical approach allows us to disentangle and quantify the ferroelectric and piezoelectric contributions separately.

We believe our current methodology provides a robust and interpretable framework, and we hope the reviewer find these clarifications helpful.

Reviewer #3

General Comment

The authors have made adequate amendments to the manuscript, and I found it ready for publication in Nature Communications.

Response

We thank the reviewer for the recommendation.

1. Reviewer #3's comment on the report from Reviewer #2

I'm aware of the contradicting inputs from Reviewer #1 and #3, but as said in my last round of review, I believe the authors have addressed all comments (that can be addressed) in their reply without getting out of scope. I think that both editors have overestimated their judgment (#1 commenting on how great the paper is and #2 about the need for modification of DFT calculations). But in my opinion, I'm leaning on the side of the reviewer #1 because the paper is mainly experimental with DFT support, while if the focus were calculations, then I would agree with the #2. In this case I think that Nature Communications is suitable journal, while more advanced calculations with quantum mechanical framework would be somewhat beneficial, I think they are not necessary as the journal is good but not as good as Nature itself for example (in which I would persist to get better microscopic origin of their reported behaviour).

So as said, I think the paper is really nice and it will be useful for the readers of Nature Communications even without additional calculations.

Response

We sincerely appreciate the reviewer's support of this work.